# Sparse Autoencoder Neural Operators: Model Recovery in Function Spaces

**Bahareh Tolooshams**[1,2,3]* **Ailsa Shen**[4]* **Anima Anandkumar**[4]
[1]University of Alberta    [2]Alberta Machine Intelligence Institute (Amii)
[3]Neuroscience and Mental Health Institute (NMHI), University of Alberta
[4] California Institute of Technology (Caltech)

## Abstract

We frame the problem of unifying representations in neural models as one of sparse model recovery and introduce a framework that extends sparse autoencoders (SAEs) to lifted spaces and infinite-dimensional function spaces, enabling mechanistic interpretability of large neural operators (NO). While the Platonic Representation Hypothesis suggests that neural networks converge to similar representations across architectures, the representational properties of neural operators remain underexplored despite their growing importance in scientific computing. We compare the inference and training dynamics of SAEs, lifted-SAE, and SAE neural operators. We highlight how lifting and operator modules introduce beneficial inductive biases, enabling faster recovery, improved recovery of smooth concepts, and robust inference across varying resolutions, a property unique to neural operators.

## 1 Introduction

A growing body of work reports that neural networks, despite architectural differences, converge to unifying representations across contexts such as brain–model comparisons [1, 2], alignment with large language models [3], brain–computer interfaces [4], and universality in dynamics [5]. To study this, methods for representation similarity have been developed [6–8], building on representational similarity analysis [9]. The Platonic Representation Hypothesis (PRH) [10] formalizes why and where such convergence arises, framing it as a unifying principle across architectures. Recent work suggests it may arise from implicit regularization in stochastic gradient descent, which favours solutions with minimal gradient norms [11]. These advances raise a central question: under what conditions do different architectures learn equivalent representations, and how can this be formalized?

If convergence is truly universal, it should also extend beyond Euclidean spaces to neural operators (NO) [12–14], which learn mappings between infinite-dimensional function spaces. Neural operators are now central to data-driven scientific modelling [15] and have recently been applied in neuroscience to capture brain representations [16, 17]. While their predictive power is well established across applications [18–20], their representational properties and interpretability remain largely unknown, leaving open whether universal representation principles extend naturally to function spaces.

To study the PRH in this broader setting, we adopt the perspective of concept learning [21], focusing on sparse model recovery [22, 23]. Rather than asking whether two models learn the same representations, we ask whether they recover the same underlying concepts, an approach aligned with mechanistic interpretability, where SAEs extract interpretable features from large networks [24–34]. By framing representation learning as estimating latent sparse codes and dictionaries, we unify comparisons across architectures, training dynamics, and inductive biases. This leads us to three central questions: i) Can recovery generalize from Euclidean to functional spaces? ii) Under what

---

*Equal contribution.

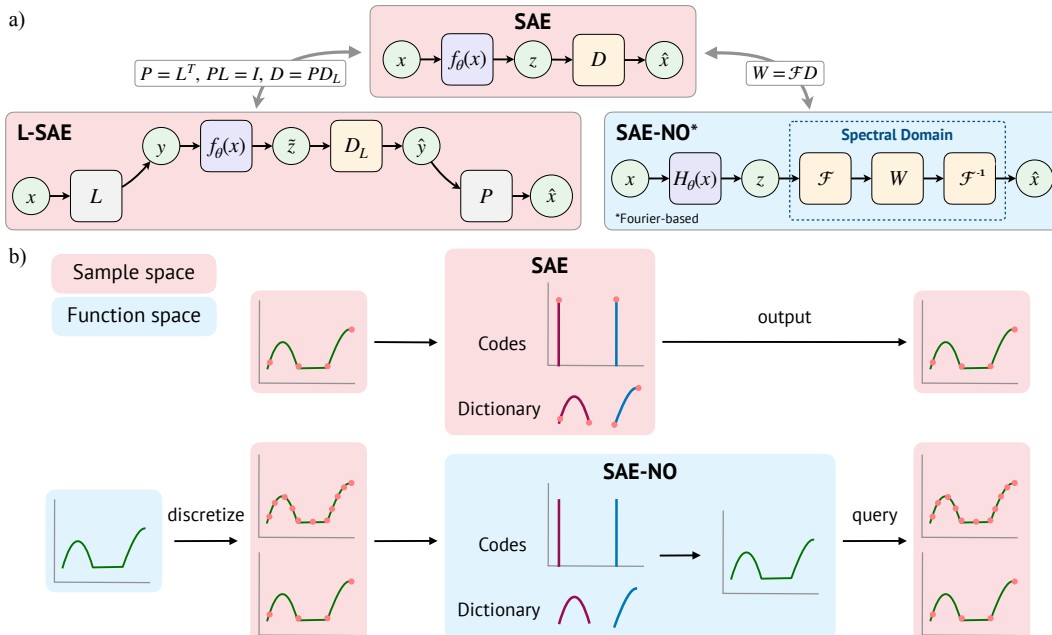

Fig. 1: **Model Recovery with SAEs.** a) Architectural comparison of SAE, lifted SAE, and SAE Neural Operators. b) Learning in sampled Euclidean spaces vs. function spaces.

conditions do networks and operators recover equivalent representations, and when do operators offer advantages? iii) How does lifting affect recovery dynamics?

**Our Contributions**  We address these questions by extending SAEs to lifted SAEs (L-SAEs), which learn concepts in a lifted space, and SAE neural operators (SAE-NOs), which formalize model recovery in function spaces. We show that lifting enables recovery while acting as a preconditioner that accelerates learning, and identify conditions where the inference and dynamics of L-SAEs reduce to those of standard SAEs. We further introduce SAE-NOs, demonstrating that i) lifting-induced preconditioning extends to L-SAE-NOs, ii) truncated modes confer an inductive bias that is advantageous for recovering smooth concepts, iii) SAE-NOs recover representations robustly under resolution changes where SAEs fail, and iv) with full-frequency filters and matched spatial support, their dynamics reduce to those of SAE-CNN.

**Preliminaries**  The sparse linear generative model (Def. 1.1), also known as sparse coding [22, 23], has a rich history in inverse problems [35], deeply studied in statistics [36] and in compressed sensing [37–40]. Representing data $x$ under a *sparse generative model* involves learning a dictionary $D^*$ to infer a sparse code $z$. We focus on the case where $D^*$ is unknown (a.k.a. *dictionary learning*) [41–43] and refer to it as *sparse model recovery*, formulated as a bilevel problem (Def. 1.2) [44].

**Definition 1.1** (Sparse Generative Models). *A data sample $x \in \mathbb{R}^m$ is said to follow a sparse generative model if there exists a sparse latent representation $z \in \mathbb{R}^p$ (with $supp(z) \leq k \ll p$) and an overcomplete dictionary $D^* \in \mathbb{R}^{m \times p}$ (with $p \gg m$) that model the data as $x = D^* z$ in the noiseless setting (see Def. B.1 for sparse convolutional generative models).*

**Definition 1.2** (Sparse Model Recovery). *Under the sparse generative model (Def. 1.1), the goal of sparse model recovery is to estimate the underlying dictionary $D^*$ by solving the following problem:*

$$\text{(outer)} \min_{D \in \mathcal{D}} \quad \frac{1}{2} \|x - Dz\|_2^2 + \lambda \mathcal{R}(z) \quad \text{s.t.} \quad \text{(inner)} \ z = f_\theta(x) = \arg\min_{v \in \mathbb{R}^p} \mathcal{L}(x, v, \theta), \tag{1}$$

*where $f_\theta$ is a forward map from data $x$ to latent code $z$, defined as the minimizer of the inner loss $\mathcal{L}(x, v, \theta)$. The outer-level optimizes $D$, and model recovery is* successful *when the learned concept $D$ approximates the true underlying dictionary $D^*$.*

The connection between bilevel optimization and neural networks is well established through SAEs [45–48]. The inner and outer levels in Def. 1.2 mirror the encoder–decoder structure of a neural network [44], where the encoder $f_\theta$ maps data $x$ to sparse codes $z$, and the decoder reconstructs $x$ via $Dz$ using learned weights $D$ (see Def. 1.3).

**Definition 1.3** (Sparse Autoencoders for Model Recovery). *The goal of model recovery with SAEs is to infer a sparse code $z \in \mathbb{R}^p$, representing the data $x \in \mathbb{R}^m$, with a linear combination of a dictionary $D$:    (encoder)   $z = f_\theta(x)$,    (decoder)   $\hat{x} = Dz$, where $\theta$ denotes the set of encoder parameters. Model recovery refers to learning the dictionary $D$ (see Def. B.1 for SAE-CNN).*

Sparse model recovery has attracted interest from two communities: a) *unrolled learning*, using optimization-inspired architectures for network design [46–54]; and b) *sparse interpretability* or mechanistic interpretability, employing SAEs to extract concepts from large models [24–34]. To extend SAEs to function spaces, we first cover basics of neural operators [12–14]. Unlike neural networks, which operate on finite-dimensional Euclidean spaces, neural operators learn mappings between infinite-dimensional function spaces over bounded domains. They have gained traction in data-driven scientific modelling [15], particularly for solving partial differential equations (PDEs), and can generalize across discretizations. Neural operators consist of the following three modules of lifting, kernel integration, and projection (see Appendix and [14]).

Fourier Neural Operator (FNO) [13] models the kernel integral operator with a convolution operator parameterized in Fourier space (Def. 1.4, see also Def. B.3 for Fourier transform). The convolution operator parameterizes the kernel $\kappa(x, y) = \kappa(x - y)$ as a complex function $\kappa : D \to \mathbb{C}^{d_w \times d_v}$ (Def. B.2). FNO is used with truncated frequency modes in practice; this has been shown to improve performance and lower sensitivity to change (decreasing) the discretization sampling [13, 14].

**Definition 1.4** (Fourier integral operator $\mathcal{K}$ (restated from [13, 14])). *Define the Fourier integral operator by $(\mathcal{K}(\phi)v)(x) = \mathcal{F}^{-1}(R_\phi \cdot (\mathcal{F}v))(x)$, $\forall x \in D$, where $R_\phi := \mathcal{F}(\kappa)$ is the Fourier transform of a periodic function $\kappa : D \to \mathbb{C}^{d_w \times d_v}$ parameterized by $\phi$.*

To extend SAEs to function spaces, we leverage the link between convolution in Euclidean space and multiplication in the Fourier domain. Building on [55] showing SAE-induced inductive biases, we investigate when networks and operators recover similar or distinct concepts. Section 2 formalizes sparse model recovery with neural operators [12–14] for concept learning in function spaces.

## 2  SAE Neural Operators for Model Recovery in Function Spaces

We extend SAEs, which are typically implemented with MLP layers (SAE-MLPs), to neural operators to enable concept recovery in function spaces. Based on operator learning, we formulate a sparse functional generative model of data (Def. 2.1), and its corresponding optimization for model recovery on function spaces (Def. 2.2).

**Definition 2.1** (Sparse Functional Generative Model). *Consider a linear operator $\mathcal{G}_{D^*} : \mathcal{Z} \to \mathcal{X}$ in function spaces, where $\mathcal{Z}$ and $\mathcal{X}$ are Banach spaces of functions defined on bounded domains $D_z \subset \mathbb{R}^p$ and $D_x \subset \mathbb{R}^m$, respectively. The data sample $x \in \mathcal{X}$ is said to follow a sparse functional generative model if there exists a sparse latent representation $z$ drawn from the support $\mathcal{Z}$ of a probability measure $\mu$, where $x = \mathcal{G}_{D^*}(z)$.*

**Definition 2.2** (Sparse Functional Model Recovery). *Under the sparse functional generative model (Def. 2.1), the goal of sparse model recovery in function spaces is to estimate the underlying operator $\mathcal{G}_{D^*}$ by solving the following bilevel optimization problem:*

$$\min_{D \in \mathcal{D}} \quad \mathbb{E}_{z \sim \mu} \|\mathcal{G}_{D^*}(z) - \mathcal{G}_D(\hat{z})\|_{\mathcal{X}}^2 \quad s.t. \quad \hat{z} = \arg\min_{v} \mathbb{E}_{z \sim \mu} \|v - \mathcal{H}_\theta(\mathcal{G}_{D^*}(z))\|_{\mathcal{Z}}^2, \quad (2)$$

*where the operator $\mathcal{H}_\theta : \mathcal{X} \to \mathcal{Z}$ aims to approximate the inverse of the operator $\mathcal{G}_{D^*}$. Model recovery is successful when the learned concepts $D$ in $\mathcal{G}_D$ closely approximate $D^*$ in $\mathcal{G}_{D^*}$.*

To recover concepts, we formalize SAE-NO (Def. 2.3) with an encoder approximating the inverse generative operator and a decoder that captures the underlying operator and its learnable concepts.

**Definition 2.3** (Sparse Autoencoder Neural Operators (SAE-NOs) for Model Recovery). *The goal of model recovery with SAE-NO in function spaces is to infer a sparse code $z$ as a sample from the function space $\mathcal{Z}$, representing the data sample from the function space $\mathcal{X}$ via the operator $\mathcal{G}_D$: (encoder)   $z = \mathcal{H}_\theta(x)$,    (decoder)   $\hat{x} = \mathcal{G}_D(z)$, where $\mathcal{H}_\theta$ is the encoder operator, and model recovery involves estimating the concepts $D$.*

We consider Fourier parametrization and define SAE Fourier Neural Operators (*SAE-FNOs*) as $\mathcal{G}_D(z) := \mathcal{F}^{-1}\big(\sum_{c=1}^C W_c \cdot (\mathcal{F}z_c)\big)$, where $W_c = \mathcal{F}D_c$ is the Fourier transform of the convolution kernel. This reduces sparse functional model recovery to recovery of a sparse convolutional generative

model (Def. B.1), implemented as a sum of convolutions between $z$ and $D$. This formulation generalizes SAEs to operator and Fourier domains, enabling concept learning in function spaces. We further extend this approach to model recovery in lifted spaces (Def. 2.4), incorporating *lifting* and *projection* modules that are crucial components of neural operators [12–14].

**Definition 2.4** (Lifted SAE-NO for Model Recovery). *The goal of model recovery with lifted SAE-NO in function spaces is to infer a sparse code $z$ as a sample from the function space $\mathcal{Z}$, representing the data from $\mathcal{X}$ via the operator $\mathcal{G}_{D_L}$:*

$$
\begin{aligned}
\text{(lifting and encoder)} \quad & z = \mathcal{H}_\theta(y), \quad y = Lx \\
\text{(decoder and projecting)} \quad & \hat{x} = P\hat{y}, \quad \hat{y} = \mathcal{G}_{D_L}(z),
\end{aligned}
\tag{3}
$$

*where $\mathcal{H}_\theta$ encodes the lifted data $y$, and model recovery is done via the learned lifted concepts $D_L$.*

Finally, we informally define *architectural inference equivalence*, i.e., when two architectures can encapsulate the same underlying model and produce the same latent representations of an input.

## 3 Results

We generate data under the sparse convolutional generative model, and study the impact of lifting and operator on model recovery on SAE-CNN, L-SAE-CNN, SAE-FNO, and L-SAE-FNO (see Appendix). The concepts in SAE-CNN (or L-SAE-CNN), SAE-FNO (or L-SAE-FNO) are parameterized as a convolution and a Fourier-based operator, respectively.

**Lifting** We examined model recovery in the lifted space and found: i) models can be recovered in the lifted space (Fig. 4); ii) lifting can act as a preconditioner (Prop. 3.1), accelerating recovery (Fig. 4) by reducing dictionary atom correlation, promoting a more isotropic loss landscape (Fig. 6); and iii) when lifting and projection are tied and orthogonal, the architectural inference (Props. 3.2 and D.1) and training dynamics (Props. 3.1 and D.2) of lifted SAEs reduces to SAEs (Fig. 5).

**Proposition 3.1** (Training Dynamics of Lifting). *The training dynamics of the lifted-SAE (L-SAE) $D_L^{(k+1)} = D_L^{(k)} + \eta_L P^\top (x - P D_L^{(k)} z) z^\top$, with lifting $L$ and projection $P$, has the effective update in the original space, expressed as: $D^{(k+1)} = D^{(k)} + \eta_L (L^\top L)(x - D^{(k)} z) z^\top$, where $L^\top L$ acts as a preconditioner, potentially accelerating learning by inducing a more isotropic update. If they satisfy the equivalent architectural inference (Prop. 3.2), then the dynamics are equivalent to those of an SAE: $D^{(k+1)} = D^{(k)} + \eta(x - D^{(k)} z) z^\top$ (see Prop. D.2 for L-SAE-CNN).*

**Proposition 3.2** (Architectural Inference Equivalence of Lifting in SAE). *The architectural inference of a SAE is equivalent to a lifted-SAE if the projection is tied to the lifting operator ($P = L^\top$), and the lifting operator is orthogonal ($L^\top L = I$), i.e., under the same learned model $D = P D_L$, the representation dynamics and output of the networks are equivalent (see Prop. D.1 for SAE-CNN and Prop. D.6 for SAE-FNO).*

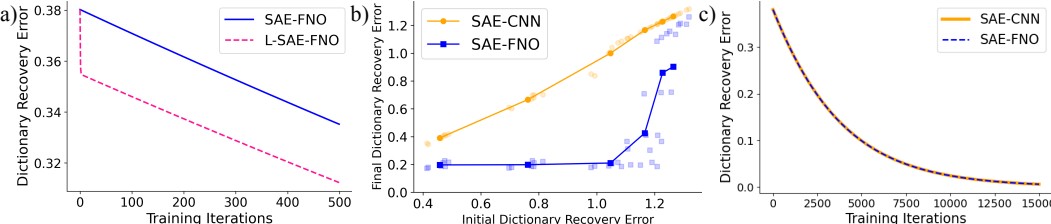

Fig. 2: **SAE-CNN vs. SAE-FNO.** a) Lifting accelerates learning. b) SAE-FNO's superiority in recovering smooth concepts via truncated Fourier modes. c) Equivalent learning when SAE-FNO uses all Fourier modes and matched spatial receptive field of SAE-CNNs.

**SAE-FNO** We examined model recovery in function spaces. Our results show that: i) the lifting-induced preconditioning effect extends to L-SAE-FNO (Fig. 2a, Prop. D.3); ii) SAE-FNO with truncated modes exhibits an inductive bias that favours recovery of smooth concepts, which is desirable when the underlying data concepts are smooth (Fig. 2b); iii) SAE-FNOs can successfully infer representations from higher-resolution data, whereas SAEs fail to recover the true underlying representation when data resolution changes (Fig. 3); and iv) when the integral operator is implemented as a convolution with weights spanning the full frequency modes and constrained to the same spatial support, the architectural inference and training dynamics of SAE-FNO (Prop. 3.3) reduce to those of SAE-CNN, with weights related by a Fourier transform (Prop. 3.4, Fig. 2c).

**Proposition 3.3** (Training Dynamics of SAE-FNO). *Consider an SAE-FNO whose integral operator is implemented as a convolution, with frequency-domain weights $\boldsymbol{W}_c$ defined as the Fourier transform of the SAE-CNN's spatial kernels, i.e., $\boldsymbol{W}_c = \mathcal{F}\boldsymbol{D}_c$. The training dynamics of the SAE-FNO are then expressed as: $\boldsymbol{D}_c^{(k+1)} = \boldsymbol{D}_c^{(k)} + \frac{\eta}{M}(\boldsymbol{x} - \frac{1}{\sqrt{M}}\mathcal{F}^{-1}[\sum_{i=1}^{C}\mathcal{F}\boldsymbol{D}_i^{(k)} \odot \mathcal{F}\boldsymbol{z}_{i,t}]) \star \boldsymbol{z}_{c,t}$ where $M$ is the full number of modes, $\odot$ denotes element-wise multiplication, and $\star$ denotes correlation. If the SAE-FNO's filters span the full frequency range and are constrained to have the same spatial support as in the SAE-CNN, then its dynamics reduce to those of the SAE-CNN (see Prop. 3.4).*

**Proposition 3.4** (Training Dynamics of SAE-CNN and SAE-FNO). *The training dynamics of the SAE-FNO's frequency-domain weights $\boldsymbol{W}_c^{(k+1)} = \boldsymbol{W}_c^{(k)} + \frac{\eta}{\sqrt{M}}\mathcal{F}\left(\boldsymbol{x} - \hat{\boldsymbol{x}}^{(k)}\right) \odot \mathcal{F}(\boldsymbol{z}_{c,t})$ has an effective update in the original space, expressed as: $\boldsymbol{D}_c'^{(k+1)} = \boldsymbol{D}_c'^{(k)} + \frac{\eta}{M}\left(\boldsymbol{x} - \hat{\boldsymbol{x}}^{(k)}\right) \star \boldsymbol{z}_{c,t}$. If the architectures satisfy the inference equivalence condition (Prop. D.4, see Prop. D.5 for lifting), then the dynamics are equivalent to that of a SAE-CNN: $\boldsymbol{D}_c^{(k+1)} = \boldsymbol{D}_c^{(k)} + \eta(\boldsymbol{x} - \sum_{i=1}^{C} \boldsymbol{D}_i^{(k)} * \boldsymbol{z}_{i,t}) \star \boldsymbol{z}_{c,t}$.*

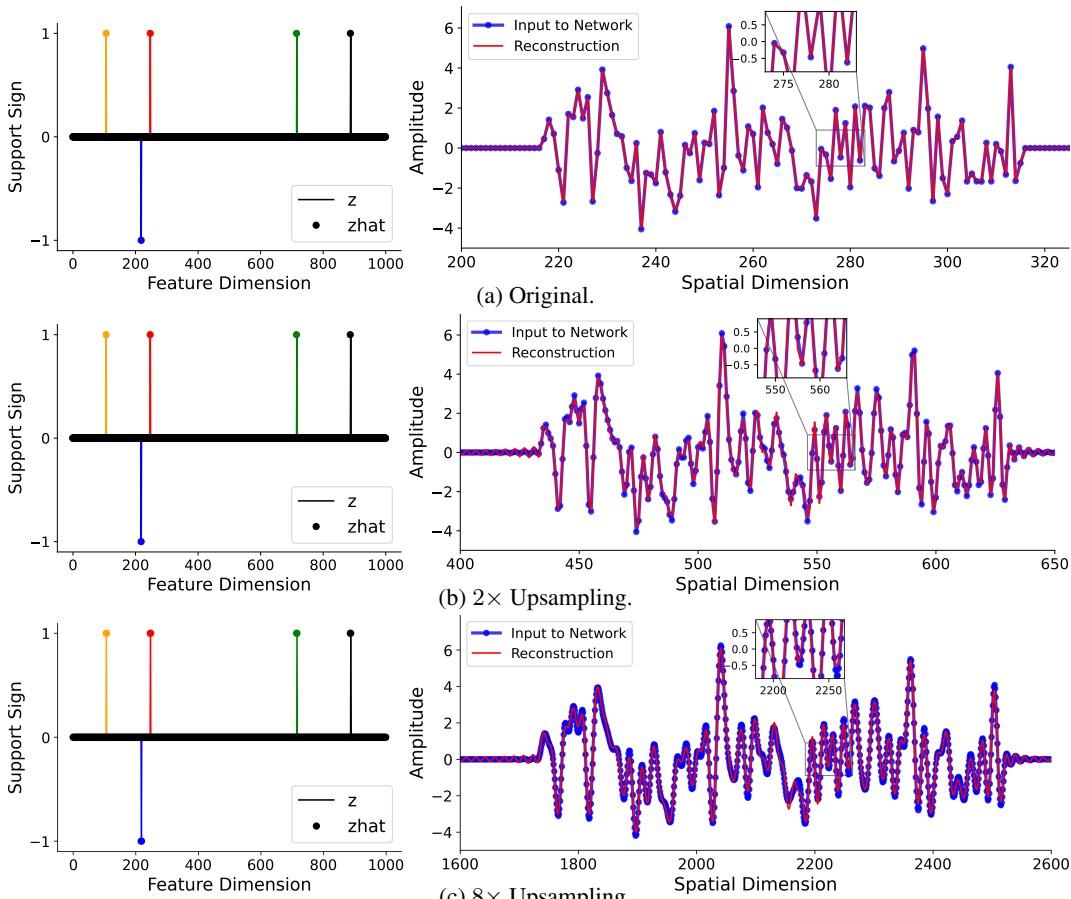

Fig. 3: **SAE-FNO Upsampling Robustness Across Resolutions.** SAE-FNO successfully infers the underlying sparse representations and reconstructs data across multiple discretization levels. The left panels show inference of 1-sparse code supports across 5 kernels, and the right panels display spatial-domain signal reconstruction (see also Fig. 7).

**Conclusion** We introduced a unified framework for sparse model recovery to analyze representations across finite- and infinite-dimensional settings. Our analysis shows that the SAE-FNO's inductive biases enable superior recovery of smooth concepts and generalization across resolutions where standard models fail. This provides a rigorous foundation for both model recovery in function spaces and mechanistic interpretability of large neural operators. While validated here on synthetic data, critical next steps include applying this framework to real-world settings, where based on the prior success of neural operators, we speculate that SAE-FNO will be highly effective in extracting meaningful concepts, including learning harmonic representations useful in scientific modelling and capturing global concepts or frequency-based coding in neuroscience.

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

# A  Appendix - Acknowledgments

A.S. conducted this work as a Dale and Suzanne Burger SURF Fellow through the Summer Undergraduate Research Fellowship (SURF) program at Caltech and gratefully acknowledges its funding. A.A. was supported by the Bren Chair Professorship and AI2050 Senior Fellowship.

B.T. proposed, designed, and led the project. A.S. implemented the methods and executed all experiments. A.S. and B.T. wrote the paper together. A.A. provided supervision, valuable feedback, and editorial comments.

# B  Appendix - Additional Background

**Notations**  We denote scalars as non-bold, lower-case a, vectors as bold-lower-case $\boldsymbol{a}$, and matrices as upper-case letters $\boldsymbol{A}$.

**Sparse model recovery**  The bi-level optimization (Def. 1.2) takes a general form. When the inner level objective $\mathcal{L}(\boldsymbol{x}, \boldsymbol{v}, \theta) = \frac{1}{2}\|\boldsymbol{x} - \boldsymbol{D}\boldsymbol{v}\|_2^2 + \mathcal{R}(\boldsymbol{v})$, this problem is reduced to the classical alternating-minimization dictionary learning [42, 43]. In this classical form, sparse coding is widely applied in science and engineering; e.g., in signal processing [56], image denoising [57], and extracting interpretable gene modules [58, 59]. When the model parameters $\boldsymbol{D}$ are known, the problem is reduced to recovering the sparse representation by solving $\min_{\boldsymbol{z} \in \mathbb{R}^p} \frac{1}{2}\|\boldsymbol{x} - \boldsymbol{D}\boldsymbol{z}\|_2^2 + \lambda\mathcal{R}(\boldsymbol{z})$, where $\mathcal{R}(\boldsymbol{z})$ is a sparse regularizer; setting $\mathcal{R}(\boldsymbol{z}) = \|\boldsymbol{z}\|_1$, the optimization problem is reduced down to lasso [60], or referred to as basis pursuit [61], solving via proximal gradient descent [62] such as iterative shrinkage-thresholding algorithm (ISTA) [63] and its fast momentum-based version [64].

**Sparse ReLU autoencoders**  For shallow ReLU autoencoders [26], the formulation reduces to

$$\min_{\boldsymbol{D} \in \mathcal{D}} \quad \frac{1}{2}\|\boldsymbol{x} - (\boldsymbol{D}\boldsymbol{z} + \boldsymbol{b}_{\text{pre}})\|_2^2 + \lambda\|\boldsymbol{z}\|_1$$
$$\text{s.t} \quad \boldsymbol{z} = \text{ReLU}(\boldsymbol{W}^\top(\boldsymbol{x} - \boldsymbol{b}_{\text{pre}}) + \boldsymbol{b}_{\text{enc}}) \tag{4}$$

This connection between bilevel optimization and neural networks, highlighted in [44, 55], has been explored extensively in the context of sparse autoencoders. Prior works have studied the gradient dynamics of variants of ReLU networks when the encoder is shallow [49, 50, 52] or deep, as in learned ISTA architectures [48]. Others have analyzed model recovery when the encoder is deep and iterative with hard-thresholding nonlinearity (JumpReLU [27]) [54].

Overall, optimizing or recovering sparse generative linear models has attracted sustained attention from two main deep learning communities over the past years; a) *Unrolling learning*: using optimization formulations to design and theoretically study neural architectures [46–54], and b) *Sparse interpretability* or mechanistic interpretability: leveraging sparse models to interpret and analyze the internal representations of complex larger networks [24–34]. We briefly expand on both works.

**Unrolled learning**  The early connections between sparse coding and deep learning trace back to sparse energy-based deep models [65, 66] and the pioneering work of LISTA [45] in constructing sparsifying recurrent neural networks. This line of work, known as unfolding [67] or unrolling [68], designs neural network layers based on iterations of an algorithm that solves an optimization problem. This connection has enabled researchers to use optimization models as a proxy to theoretically study accelerated convergence [51, 46], gradient dynamics [50, 52, 49, 53], and model recovery in both shallow [50] and deep neural networks [54, 48]. Furthermore, several works have explored the theoretical connections between convolutional neural networks and convolutional sparse coding [69, 70], or more generally, between deeply sparse signal representations and deep neural networks [71]. An example of a deep unrolled JumpReLU neural network for sparse model recovery can be formulated as the following bilevel optimization problem.

$$\min_{\boldsymbol{D} \in \mathcal{D}} \quad \|\boldsymbol{x} - \boldsymbol{D}\boldsymbol{z}_T\|_2^2 \quad \text{s.t.} \quad \boldsymbol{z}_t = \text{JumpReLU}_\lambda(\boldsymbol{z}_{t-1} - \alpha\boldsymbol{D}^\top(\boldsymbol{D}\boldsymbol{z}_{t-1} - \boldsymbol{x})) \tag{5}$$

for $t = 1, \ldots, T$, where $\boldsymbol{z}_0 = \boldsymbol{0}$, $\alpha$ is the step size, and $\text{JumpReLU}_\lambda(\boldsymbol{z}) = \boldsymbol{z} \cdot \mathbb{1}_{\boldsymbol{z} > \lambda}$, where $\mathbb{1}$ is the indicator function, and $\lambda$ controls the sparsity level of the representation. The inner mapping is now a deep/iterative encoder, which for simplicity we denote by $\boldsymbol{z}_T = f_\theta(\boldsymbol{x})$. The outer objective enforces the structure of the decoder and the loss function. The bilevel optimization in eq. (5) can be mapped to the following neural network autoencoder architecture, which uses a recurrent and residual encoder.

The synthetic experiments in this paper use this encoder in its architecture for sparse model recovery.

$$\text{(encoder)} \quad \boldsymbol{z}_t = \text{JumpReLU}_\lambda(\boldsymbol{z}_{t-1} - \alpha \boldsymbol{D}^\top(\boldsymbol{D}\boldsymbol{z}_{t-1} - \boldsymbol{x})), \quad \text{for} \quad t = 1, \ldots, T$$
$$\text{(decoder)} \quad \hat{\boldsymbol{x}} = \boldsymbol{D}\boldsymbol{z}_T. \tag{6}$$

For experiments recovering the sparse convolutional generative model, we use the architecture that takes convolution blocks, as shown below.

$$\text{(encoder)} \quad \boldsymbol{z}_{c,t} = \text{JumpReLU}_\theta(\boldsymbol{z}_{c,t-1} - \alpha \boldsymbol{D}_c \star (\sum_{c=1}^{C} \boldsymbol{D}_c * \boldsymbol{z}_{c,t-1} - \boldsymbol{x})), \quad \text{for} \quad t = 1, \ldots, T$$

$$\text{(decoder)} \quad \hat{\boldsymbol{x}} = \boldsymbol{D} * \boldsymbol{z}_T$$

$$\tag{7}$$

for $t = 1, \ldots, T$ and $c = 1, \ldots, C$, where $\boldsymbol{z}_0 = \boldsymbol{0}$, $\alpha$ is the step size, $*$ is convolution operator, and $\star$ is a correlation operator.

**Sparse Interpretability** Interpretability in deep learning [72], particularly through extracting concepts learned by deep neural networks [21], has gained prominence in the era of large models. Motivated by the linear representation hypothesis (LRH) [24, 28], sparse autoencoders (SAEs) are widely used to recover interpretable structures from internal representations of large language models [25–27, 31] and vision models [32]. Recent studies show that the architecture of SAEs imposes inductive biases on which concepts are recoverable [55]. Building on this, we investigate conditions under which neural networks and neural operators recover similar concepts, and we explore their respective inductive biases. Our formulation of sparse model recovery in function spaces enables the use of SAEs operating on functions to interpret large neural operators.

**Convolutional Setting** Sparse generative models can be extended to *sparse convolutional genera-tive models* (Def. B.1), where the concepts are locally and sparsely appearing in the data.

**Definition B.1** (Sparse Convolutional Generative Models). *A data example* $\boldsymbol{x} \in \mathbb{R}^m$ *is said to follow a sparse convolutional generative model if there exists a sparse latent representation* $\boldsymbol{z} = \{\boldsymbol{z}_c \in \mathbb{R}^p\}_{c=1}^{C}$ *(with* $\text{supp}(\boldsymbol{z}_c) \le k \ll p$*) and a set of $C$ localized dictionary* $\{\boldsymbol{D}_c^* \in \mathbb{R}^h\}_{c=1}^{C}$ *(with* $h \ll m$*) such that* $\boldsymbol{x} = \sum_{c=1}^{C} \boldsymbol{D}_c^* * \boldsymbol{z}_c$ *in the noiseless setting, where* $*$ *denotes the convolution operator.*

**Neural operators** Neural operators consist of the following three modules:

- **Lifting**: This is a fully local operator which we model by a matrix $\boldsymbol{L} : \mathbb{R}^m \to \mathbb{R}^{d_{v_0}}$. It maps the input $\{\boldsymbol{x} : D_x \to \mathbb{R}^m\}$ into a latent representation $\{\boldsymbol{v}_0 : D_0 \to \mathbb{R}^{d_{v_0}}\}$, where $h > m$.
- **Kernel Integration**: For $t = 0, \ldots, T - 1$, this is a non-local integral kernel operator that maps representation at one layer $\{\boldsymbol{v}_t : D_t \to \mathbb{R}^{d_{v_t}}\}$ to the next $\{\boldsymbol{v}_{t+1} : D_{t+1} \to \mathbb{R}^{d_{v_{t+1}}}\}$ (see Def. B.2).
- **Projection**: This is a fully local operator, similar to the lifting operator. It maps the filtered lifted data to the output function, i.e., $\boldsymbol{P} : \mathbb{R}^{d_{v_T}} \to \mathbb{R}^m$, where $d_{v_T} > m$.

where the kernel integral operator is used to map $\boldsymbol{x}$ to an estimate of the representation $\boldsymbol{z}$, the input/output domains would be $D_x$ and $D_z$, respectively. Moreover, when the kernel integral operator is used to refine the latent representation $\boldsymbol{z}$ from one layer to another, the input/output domain of the operator would be both $D_z$.

**Definition B.2** (Kernel integral operator $\mathcal{K}$ (restated from [13, 14])). *Define the kernel integral operator by*

$$(\mathcal{K}_t(\boldsymbol{v}_t))(x) := \int_{D_t} \kappa^{(t)}(x, y)\boldsymbol{v}_t(y)d\nu_t(y), \quad \forall x \in D_{t+1} \tag{8}$$

*where* $\kappa^{(t)} : C(D_{t+1} \times D_t; \mathbb{R}^{d\nu_{t+1} \times d\nu_t})$ *are the parameters of the kernels, modeled by a neural network, and $\nu_t$ is a Borel measure on $D_t$, where $C$ denotes the space of continuous functions.*

**Definition B.3** (Fourier transform). *Let* $\mathcal{F}$ *be the Fourier transform of the function* $\boldsymbol{v} : D \to \mathbb{R}^{d_v}$, *whose inverse is denoted by* $\mathcal{F}^{-1}$ *on the function* $\boldsymbol{w} : D \to \mathbb{C}^{d_w}$. *We have*

$$(\mathcal{F}\boldsymbol{v})_j(k) = \int_D \boldsymbol{v}_j(x)e^{-2i\pi\langle x, k\rangle}dx, \qquad j = 1, \ldots, d_v$$

$$(\mathcal{F}^{-1}\boldsymbol{w})_j(x) = \int_D \boldsymbol{w}_j(k)e^{2i\pi\langle x, k\rangle}dk, \qquad j = 1, \ldots, d_w \tag{9}$$

*where* $i = \sqrt{-1}$ *is the imaginary unit.*

# C   Appendix - Additional Experimental Results

## C.1   Experimental Setup

**Sparse Generative Model**    The results shown in Fig. 6 are from a synthetic dataset that follows a sparse generative model (Def. 1.1). The atoms of the ground-truth dictionary $D^* \in \mathbb{R}^{1000 \times 1500}$ are drawn from a standard normal distribution and then $\ell_2$-normalized. The dataset consists of 50,000 samples $x \in \mathbb{R}^{1000}$. Each latent code $z \in \mathbb{R}^{1500}$ has a total sparsity of 20. The amplitudes of the non-zero elements are drawn from a sub-Gaussian distribution $\mathcal{N}(15, 1)$.

**SAE**    For SAE (SAE-MLP) and L-SAE (L-SAE-MLP) models, the encoder implements $f_\theta(x)$ following Def. 1.2 using a deep unrolled JumpReLU network (eqs. (5) and (6)) with $T = 50$ layers, as described in the sparse model recovery framework (Def. 1.2). The non-linearity is Hard-Thresholding with a threshold of $\lambda = 0.5$, and the algorithm's internal step-size is $\alpha = 0.2$. The decoder reconstructs data as $\hat{x} = Dz$ following Def. 1.3. The dictionary weights for SAE and L-SAE are initialized with a noisy ($\sigma = 0.02$) version of the ground-truth dictionary. For the L-SAE, the 1000-dimensional input is lifted to a 1200-dimensional space. Both models are trained to minimize the Mean Squared Error (MSE) loss using the ADAM optimizer with a learning rate of $\eta = 10^{-3}$.

**Sparse Convolutional Generative Model**    The results shown in Fig. 2 are from a synthetic dataset that follows a sparse convolutional generative model (Def. B.1). The ground-truth dictionary $D^*$ consists of $p = 5$ kernels with spatial support $h = 99$. Each kernel is a multi-channel signal with 64 input channels. The values for each kernel are drawn from a standard normal distribution and then normalized. The dataset consists of 50,000 samples $x \in \mathbb{R}^{64 \times 1000}$. Each sample is generated by convolving a single (i.e., sparsity of 1) randomly chosen kernel from the dictionary with a sparse feature map, whose non-zero amplitudes are drawn from a sub-Gaussian distribution.

**SAE-CNN**    For SAE-CNN and L-SAE-CNN models, the encoder implements the convolutional version of the deep unrolled network (eq. (7)) with $T = 50$ layers, as described in the sparse model recovery framework (Def. 1.2). The non-linearity is Hard-Thresholding with a threshold of $\lambda = 10$, and the algorithm's internal step-size is $\alpha = 0.01$. The dictionary for both SAE-CNN and L-SAE-CNN are initialized with a noisy ($\sigma = 0.05$) version of the ground-truth kernels. For L-SAE-CNN, the 64-channel input is lifted to a 128-dimensional space. Both models are trained to minimize the Mean Squared Error (MSE) loss using the SGD optimizer with a learning rate of $\eta = 0.04$.

**SAE-FNO**    For SAE-FNO and L-SAE-FNO models, the encoder implements a deep unrolled network with $T = 50$ layers, as described in the sparse functional model recovery framework (Def. 2.2). The non-linearity is Hard-Thresholding with a threshold of $\lambda = 10$, and the algorithm's internal step-size is $\alpha = 0.01$. The dictionary kernels are initialized with a noisy ($\sigma = 0.05$) version of the ground-truth. For L-SAE-FNO, the 64-channel input is lifted to a 128-dimensional space. Both SAE-FNO and L-SAE-FNO are trained to minimize the Mean Squared Error (MSE) loss using the SGD optimizer with a learning rate of $\eta_L = 20.04$. All experiments are based on these settings, except Fig. 2b and Fig. 3.

Fig. 2b uses a modified experimental setup from the sparse convolutional generative model described above. The data consists of single channel signal with 1-sparse codes ($p = 1$ kernel), where the amplitudes of non-zero elements are drawn from a sub-Gaussian distribution $\mathcal{N}(5, 1)$. Training uses threshold $\lambda = 0.2$, step size $\alpha = 0.05$, and ADAM optimizer with learning rates $2 \times 10^{-5}$ (SAE-CNN) and 0.01 (SAE-FNO).

Fig. 3 uses the experimental setup of Sparse Convolutional Generative Model with $1\times, 2\times, 4\times$, and $8\times$ upsampling rates. The upsampling process is as follows: original low-resolution data ($m = 1000$) is first upsampled by inserting zeros at intervals corresponding to the upsampling rate (e.g., for $4\times$ upsampling, 3 zeros are inserted between every original sample, creating a signal of length 4000). This process introduces high-frequency spectral replicas of the original signal. The expanded signal is then processed through a 10th-order Butterworth low-pass filter using zero-phase filtering (filtfilt) to remove aliasing and ensure smooth interpolation.

## C.2   Experimental Results

Additional experimental results are shown in Figs. 4 to 7.

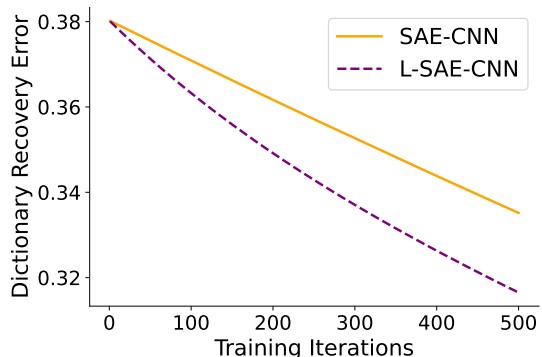

Fig. 4: **Lifting as a preconditioner.** Lifting accelerates learning.

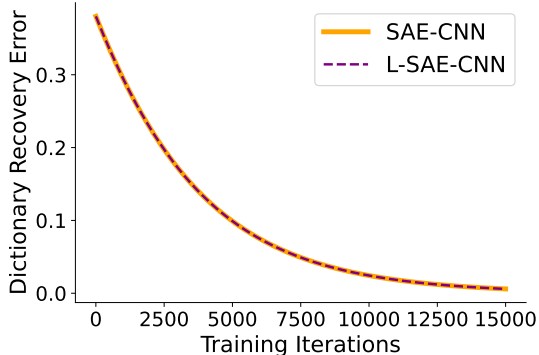

Fig. 5: **Lifting.** When the lifting operator satisfies the orthogonal condition $L^\top L = I$, the lifted SAE-CNN (L-SAE-CNN) exhibits equivalent learning dynamics to the SAE-CNN .

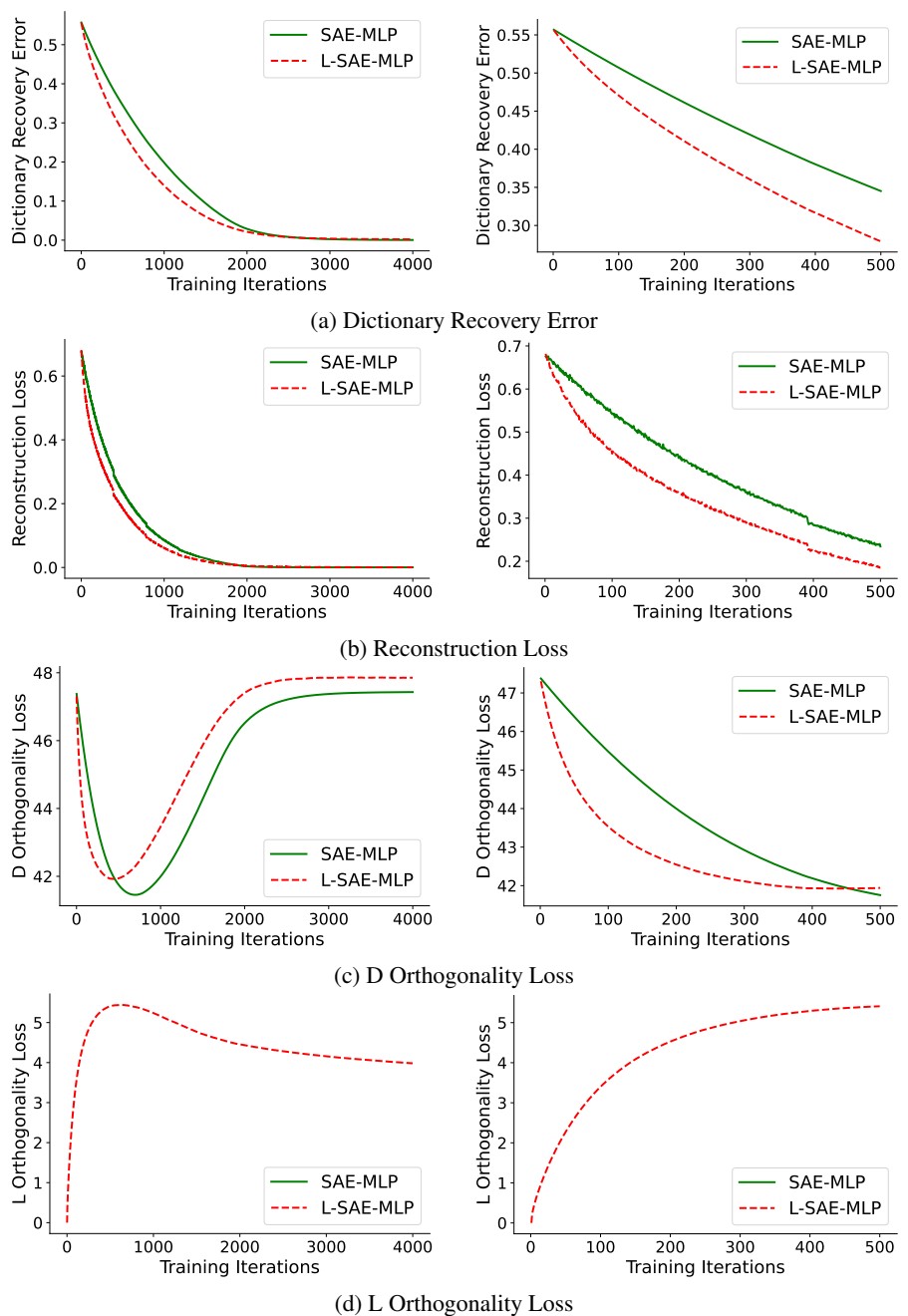

Fig. 6: **SAEs vs. L-SAEs.** Learning the lifting operator $L$ accelerates model recovery. (a) Dictionary recovery error converges faster for L-SAE-MLP, confirming the preconditioning effect of lifting; (b) Reconstruction loss follows similar convergence trends as dictionary recovery error; (c) Lifting encourages the effective dictionary $D$ to learn more orthogonal (less correlated) atoms early in training, creating a more isotropic loss landscape that accelerates recovery; (d) The lifting operator $L$ initially becomes less orthogonal, allowing the dictionary $D$ to become more orthogonal.

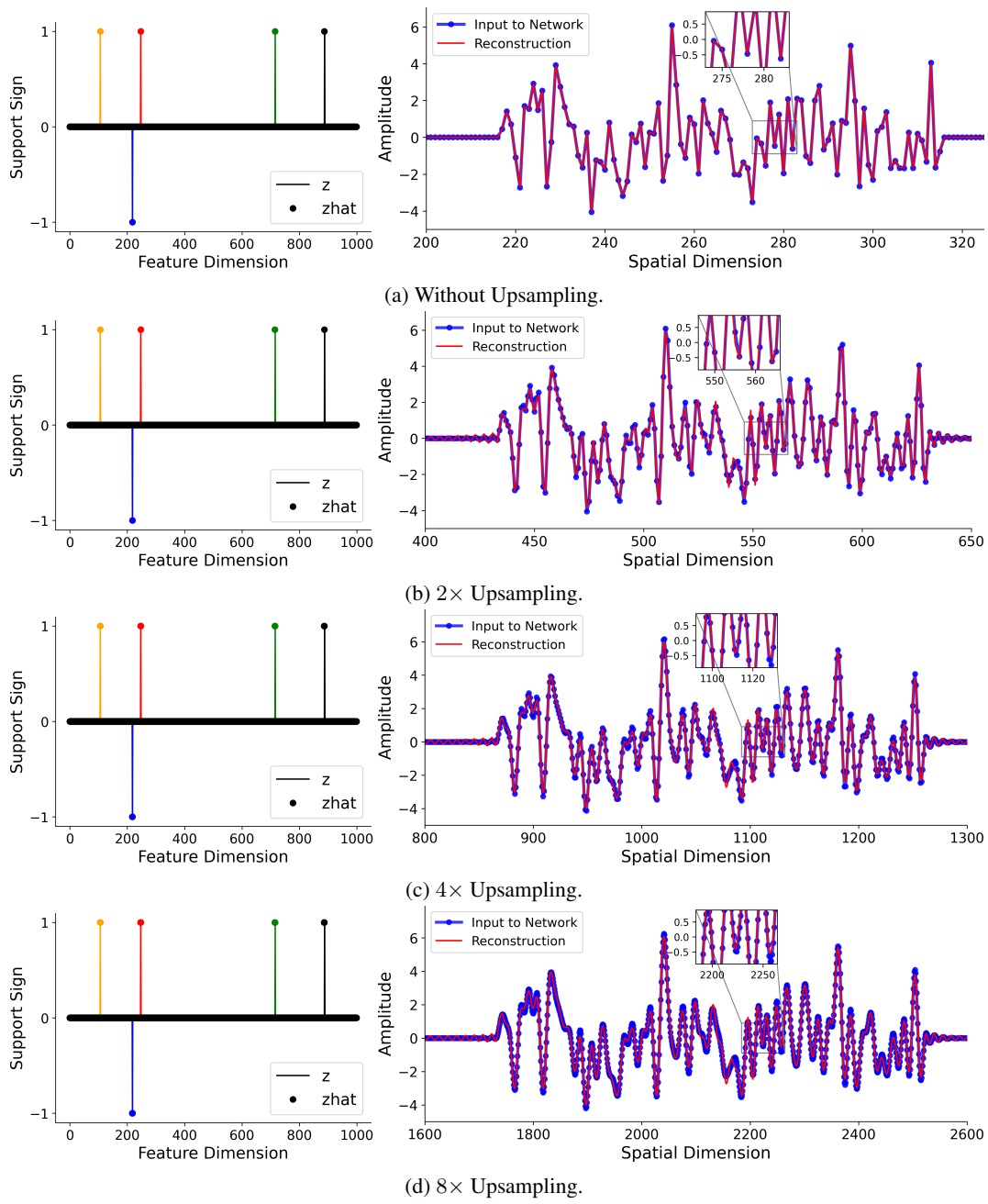

(a) Without Upsampling.

(b) 2× Upsampling.

(c) 4× Upsampling.

(d) 8× Upsampling.

Fig. 7: **SAE-FNO Upsampling Robustness Across Resolutions.** SAE-FNO successfully infers the underlying sparse representations and reconstructs data across multiple discretization levels. The left panels show inference of 1-sparse code supports across 5 kernels, while the right panels display spatial-domain signal reconstruction. (a) Original resolution ($1\times$): Baseline performance at training resolution. (b-d) Higher upsampling rates ($2\times, 4\times, 8\times$): SAE-FNO maintains accurate code support inference and reconstruction as spatial resolution increases. Across all resolutions, the inferred support remains consistent with the ground truth, and reconstructions closely match the input signals. This resolution invariance property is unique to neural operators and represents a significant practical advantage over SAE-CNNs, which typically fail when input resolution differs from their training resolution. The function space of neural operators enables this generalization across discretizations.

# D   Appendix - Theoretical Analysis

This section provides the mathematical derivations for the training dynamics of the architectures discussed in the main text. We derive the gradient descent update rules for a single data sample, which are used to learn the dictionary parameters. These derivations consider the *analytic gradient*, i.e., the dictionary gradient is computed given the codes [47, 48].

**Proposition 3.1** (Training Dynamics of Lifting). *The training dynamics of the lifted-SAE (L-SAE)* $D_L^{(k+1)} = D_L^{(k)} + \eta_L P^\top (x - P D_L^{(k)} z) z^\top$*, with lifting $L$ and projection $P$, has the effective update in the original space, expressed as:* $D^{(k+1)} = D^{(k)} + \eta_L (L^\top L)(x - D^{(k)} z) z^\top$*, where $L^\top L$ acts as a preconditioner, potentially accelerating learning by inducing a more isotropic update. If they satisfy the equivalent architectural inference (Prop. 3.2), then the dynamics are equivalent to those of an SAE:* $D^{(k+1)} = D^{(k)} + \eta(x - D^{(k)} z) z^\top$ *(see Prop. D.2 for L-SAE-CNN).*

*Proof.* For the SAE, we learn a dictionary $D$ by minimizing the reconstruction loss for the data sample $x$. The loss function is:

$$\mathcal{L}(D) = \frac{1}{2} \|x - Dz\|_2^2 \tag{10}$$

We consider the case of SAE-MLP. The gradient of the loss for $D$ is:

$$\frac{\partial \mathcal{L}}{\partial D} = -(x - Dz)z^\top \tag{11}$$

The analytic gradient update, given $z$, for $D$ at iteration $k$ with a learning rate $\eta$ is therefore:

$$D^{(k+1)} = D^{(k)} - \eta \frac{\partial \mathcal{L}}{\partial D^{(k)}} = D^{(k)} + \eta(x - D^{(k)} z)z^\top \tag{12}$$

For lifted-SAE, we learn a dictionary $D_L \in \mathbb{R}^{h \times p}$ in a higher-dimensional lifted space. The input $x$ is lifted by $L \in \mathbb{R}^{h \times m}$ and the reconstruction is projected back by $P \in \mathbb{R}^{m \times h}$.

The loss function is defined in the original data space:

$$\mathcal{L}(D) = \frac{1}{2} \|x - P D_L z\|_2^2 \tag{13}$$

The analytic gradient given $z$ of the loss with respect to $D_L$ is:

$$\frac{\partial \mathcal{L}}{\partial D} = -P^\top (x - P D_L z)z^\top \tag{14}$$

The gradient update rule for the lifted dictionary $D_L$ is:

$$D_L^{(k+1)} = D_L^{(k)} - \eta_L \frac{\partial \mathcal{L}}{\partial D_L^{(k)}} = D_L^{(k)} + \eta_L P^\top (x - P D_L z)(z)^\top \tag{15}$$

To understand the learning dynamics in the original space, we first incorporate the underlying model from L-SAEs into SAE using the relation $D' = P D_L$. Then, we derive the update rule for the effective dictionary, $D'$. Left-multiplying the update for $D_L^{(k+1)}$ by $P$, we have:

$$D'^{(k+1)} = D'^{(k)} + \eta_L (P P^\top)(x - D'^{(k)} z)(z)^\top \tag{16}$$

In our specific architecture, we constrain the projection matrix to be the transpose of the lifting matrix such that $P = L^\top$. Substituting this into the update rules yields:

$$D_L^{(k+1)} = D_L^{(k)} + \eta_L L(x - L^\top D_L z)(z)^\top \tag{17}$$

$$D'^{(k+1)} = D'^{(k)} + \eta_L (L^\top L)(x - D'^{(k)} z)(z)^\top \tag{18}$$

Hence, when $L^\top L = I$, the lifted-SAE shows equivalent learning dynamics as the SAE for model recovery. ∎

**Proposition 3.2** (Architectural Inference Equivalence of Lifting in SAE). *The architectural inference of a SAE is equivalent to a lifted-SAE if the projection is tied to the lifting operator ($P = L^\top$), and the lifting operator is orthogonal ($L^\top L = I$), i.e., under the same learned model $D = PD_L$, the representation dynamics and output of the networks are equivalent (see Prop. D.1 for SAE-CNN and Prop. D.6 for SAE-FNO).*

*Proof.* We provide the proof for the sparse generative model case when the encoder implements a proximal gradient descent type of operation in an unrolled learning setting (e.g., 6). For the SAE, the representation refinement at every encoder layer follows:

$$z_{t+1} = \mathcal{R}(z_t - \alpha D^\top (Dz_t - x)) \tag{19}$$

By construction, we incorporate the underlying model from the lifted-SAE to the standard SAE: $D = PD_L$, and re-write the encoder as:

$$z_{t+1} = \mathcal{R}(z_t - \alpha(PD_L)^\top (PD_L z_t - x)) = \mathcal{R}(z_t - \alpha D_L^\top P^\top (PD_L z_t - x)) \tag{20}$$

Using the assumption on the lifting-projection,

$$z_{t+1} = \mathcal{R}\left(z_t - \alpha(D_L)^\top (D_L z_t - Lx)\right) \tag{21}$$

This final expression is the iterative encoder update for $z$ in the lifted space, where the dictionary is $D_L$ and the input is lifted by $L$. This completes the proof of architectural inference equivalence on the standard and lifted architectures. ∎

**Proposition D.1** (Architectural Inference Equivalence of Lifting in SAE-CNN). *The architectural inference of a SAE-CNN is equivalent to a lifted SAE-CNN (L-SAE-CNN) if the projection is tied to the lifting operator ($P = L^\top$) that is orthogonal ($L^\top L = I$), i.e., under the same learned model $D_c = PD_{L,c}$, the representation dynamics and output of the networks are equivalent.*

*Proof.* We provide the proof for the sparse convolutional generative model case when the encoder implements a proximal gradient descent type of operation in an unrolled learning setting (e.g., 7). For the SAE-CNN, the representation refinement for the $c$-th feature map at every encoder layer follows:

$$z_{c,t+1} = \mathcal{R}\left(z_{c,t} - \alpha \left(\sum_{i=1}^{C} D_i * z_{i,t} - x\right) \star D_c\right) \tag{22}$$

By construction, we incorporate the underlying model from the lifted-SAE-CNN to the standard SAE-CNN: $D_c^{(k)} = PD_{L,c}^{(k)}$, and re-write the encoder using the adjoint property:

$$z_{c,t+1} = \mathcal{R}\left(z_{c,t} - \alpha \left(P^\top \left[\left(\sum_{i=1}^{C} PD_{L,i} * z_{i,t}\right) - x\right] \star D_{L,c}\right)\right) \tag{23}$$

$$z_{c,t+1} = \mathcal{R}\left(z_{c,t} - \alpha \left(\sum_{i=1}^{C} (P^\top P)D_{L,i} * z_{i,t} - P^\top x\right) \star D_{L,c}\right) \tag{24}$$

Using the assumption on the lifting-projection,

$$z_{c,t+1} = \mathcal{R}\left(z_{c,t} - \alpha \left(\sum_{i=1}^{C} D_{L,i} * z_{i,t} - Lx\right) \star D_{L,c}\right) \tag{25}$$

This final expression is the iterative encoder update for $z_c$ in the lifted space, where the dictionary is $D_L$ and the input is lifted by $L$. This completes the proof of architectural inference equivalence for SAE-CNN and L-SAE-CNN. ∎

**Proposition D.2.** *[Training Dynamics of Lifted SAE-CNN] The training dynamics of the lifted-SAE-CNN (L-SAE-CNN) $D_{L,c}^{(k+1)} = D_{L,c}^{(k)} + \eta_L L \left(x - L^\top \left(\sum_{i=1}^{C} D_{L,i}^{(k)} * z_{i,t}\right)\right) \star z_{c,t}$, with a lifting operator $L$ and projection $P$, has the effective update in the original space, expressed as: $D_c^{(k+1)} =$*

$\boldsymbol{D}_c^{(k)} + \eta_L(\boldsymbol{L}^\top \boldsymbol{L})\left(\boldsymbol{x} - \sum_{i=1}^{C} \boldsymbol{D}_i^{(k)} * \boldsymbol{z}_{i,t}\right) \star \boldsymbol{z}_{c,t}$, *where $\boldsymbol{L}^\top \boldsymbol{L}$ acts as a preconditioner, potentially accelerating learning by inducing a more isotropic update. If the architectures satisfy the architectural inference equivalence condition (Prop. D.1), then the dynamics become equivalent to that of a SAE-CNN:* $\boldsymbol{D}_c^{(k+1)} = \boldsymbol{D}_c^{(k)} + \eta\left(\boldsymbol{x} - \sum_{i=1}^{C} \boldsymbol{D}_i^{(k)} * \boldsymbol{z}_{i,t}\right) \star \boldsymbol{z}_{c,t}$.

*Proof.* For the SAE-CNN, we learn a dictionary $\boldsymbol{D}$ by minimizing the reconstruction loss for the data sample $\boldsymbol{x}$. The loss function is:

$$\mathcal{L}(\boldsymbol{D}) = \frac{1}{2}\left\|\boldsymbol{x} - \sum_{c=1}^{C} \boldsymbol{D}_c * \boldsymbol{z}_c\right\|_2^2 \tag{26}$$

The gradient of the loss with respect to the $c$-th kernel $\boldsymbol{D}_c$ is:

$$\frac{\partial \mathcal{L}}{\partial \boldsymbol{D}_c} = -\left(\boldsymbol{x} - \sum_{i=1}^{C} \boldsymbol{D}_i * \boldsymbol{z}_i\right) \star \boldsymbol{z}_c \tag{27}$$

where $\star$ denotes cross-correlation.

The analytic gradient update, given $\boldsymbol{z}$, for $\boldsymbol{D}_c$ at iteration $k$ with a learning rate $\eta$ is:

$$\boldsymbol{D}_c^{(k+1)} = \boldsymbol{D}_c^{(k)} + \eta\left(\boldsymbol{x} - \sum_{i=1}^{C} \boldsymbol{D}_i^{(k)} * \boldsymbol{z}_{i,t}\right) \star \boldsymbol{z}_{c,t} \tag{28}$$

For L-SAE-CNN, we learn a dictionary $\boldsymbol{D}_L = \{\boldsymbol{D}_{L,c}\}_{c=1}^{C}$ that operates in a higher-dimensional lifted space. The input signal $\boldsymbol{x}$ is first lifted by $\boldsymbol{L}$ (acts on the channel dimension) to $\tilde{x}$ and the reconstruction $\hat{\boldsymbol{x}}$ is obtained by projecting the lifted reconstruction $\hat{\tilde{x}} = \sum_c \boldsymbol{D}_{L,c} * \boldsymbol{z}_c$ back by $\boldsymbol{P}$.

The loss function is defined in the original data space:

$$\mathcal{L}(\boldsymbol{D}) = \frac{1}{2}\left\|\boldsymbol{x} - \boldsymbol{P}\left(\sum_{c=1}^{C} \boldsymbol{D}_{L,c} * \boldsymbol{z}_c\right)\right\|_2^2 \tag{29}$$

The analytic gradient of the loss, given $\boldsymbol{z}$, with respect the $c$-th lifted kernel $\boldsymbol{D}_{L,c}$ is:

$$\frac{\partial \mathcal{L}}{\partial \boldsymbol{D}_{L,c}} = -\boldsymbol{P}^\top\left(\boldsymbol{x} - \boldsymbol{P}\left(\sum_{i=1}^{C} \boldsymbol{D}_{L,i} * \boldsymbol{z}_i\right)\right) \star \boldsymbol{z}_c \tag{30}$$

The gradient update rule for $\boldsymbol{D}_{L,c}$ is:

$$\boldsymbol{D}_{L,c}^{(k+1)} = \boldsymbol{D}_{L,c}^{(k)} + \eta_L \boldsymbol{P}^\top\left(\boldsymbol{x} - \boldsymbol{P}\left(\sum_{i=1}^{C} \boldsymbol{D}_{L,i}^{(k)} * \boldsymbol{z}_{i,t}\right)\right) \star \boldsymbol{z}_{c,t} \tag{31}$$

To understand the learning dynamics in the original space, we derive the update rule for the effective dictionary $\boldsymbol{D}_c' = \boldsymbol{P}\boldsymbol{D}_{L,c}$, i.e., the case where both frameworks contain the same underlying model but in a different form. Left-multiplying the update for $\boldsymbol{D}_{L,c}^{(k+1)}$ update rule by $\boldsymbol{P}$, we get:

$$\boldsymbol{D}_c'^{(k+1)} = \boldsymbol{D}_c'^{(k)} + \eta_L(\boldsymbol{P}\boldsymbol{P}^\top)\left(\boldsymbol{x} - \sum_{i=1}^{C} \boldsymbol{D}_i'^{(k)} * \boldsymbol{z}_{i,t}\right) \star \boldsymbol{z}_{c,t} \tag{32}$$

In our specific architecture, we constrain the projection matrix to be the transpose of the lifting matrix such that $\boldsymbol{P} = \boldsymbol{L}^\top$. Substituting this into the update rules yields:

$$\boldsymbol{D}_{L,c}^{(k+1)} = \boldsymbol{D}_{L,c}^{(k)} + \eta_L \boldsymbol{L}\left(\boldsymbol{x} - \boldsymbol{L}^\top\left(\sum_{i=1}^{C} \boldsymbol{D}_{L,i}^{(k)} * \boldsymbol{z}_{i,t}\right)\right) \star \boldsymbol{z}_{c,t} \tag{33}$$

$$D_c'^{(k+1)} = D_c'^{(k)} + \eta_L(L^\top L)\left(x - \sum_{i=1}^{C} D_i'^{(k)} * z_{i,t}\right) \star z_{c,t} \tag{34}$$

Hence, when $L^\top L = I$, the lifted-SAE-CNN shows equivalent learning dynamics as the standard SAE-CNN for model recovery. ∎

**Proposition D.3.** *[Training Dynamics of Lifted SAE-FNO] The training dynamics of the Lifted-SAE-FNO (L-SAE-FNO)'s frequency domain weights* $W_{L,c}^{(k+1)} = W_{L,c}^{(k)} + \frac{\eta_L}{\sqrt{M}}\mathcal{F}\left(L\left(x - \hat{x}^{(k)}\right)\right) \odot \mathcal{F}(z_{c,t})$, *with lifting operator* $L$ *and projection* $P$, *has an effective update in the original space, expressed as:* $D_c^{(k+1)} = D_c^{(k)} + \frac{\eta_L}{M}\left((L^\top L)\left(x - \hat{x}^{(k)}\right)\right) \star z_{c,t}$, *where* $L^\top L$ *acts as a preconditioner, potentially accelerating learning by inducing a more isotropic update. If the architectures satisfy the architectural inference equivalence condition ([Prop. D.6](#)), then the dynamics are equivalent to that of an SAE-FNO:* $D_c'^{(k+1)} = D_c'^{(k)} + \frac{\eta}{M}\left(x - \frac{1}{\sqrt{M}}\mathcal{F}^{-1}\left(\sum_{i=1}^{C}\mathcal{F}(D_i^{(k)}) \odot \mathcal{F}(z_{i,t})\right)\right) \star z_{c,t}$.

*Proof.* For the SAE-FNO, we learn frequency-domain weights $W_c = \mathcal{F}D_c$ by minimizing the reconstruction loss for the data sample $x$. Taking normalization into account, the reconstruction operator is $\hat{x} = \mathcal{G}_W(z) = \frac{1}{\sqrt{M}}\mathcal{F}^{-1}\left(\sum_{c=1}^{C} W_c \cdot (\mathcal{F}z_c)\right)$, where $W_c = \mathcal{F}D_c$, where $M$ is the number of modes. The loss function is:

$$\mathcal{L}(W) = \frac{1}{2}\mathbb{E}_{z\sim\mu}\left[\|x - \mathcal{G}_W(z)\|_{\mathcal{X}}^2\right] \tag{35}$$

where $x = \mathcal{G}_{D^*}(z)$ is the ground-truth signal.

The gradient of the loss with respect to the $c$-th frequency-domain kernel $W_c$ is:

$$\frac{\partial\mathcal{L}}{\partial W_c} = -\frac{1}{\sqrt{M}}\mathcal{F}(x - \hat{x}) \odot \mathcal{F}(z_c) \tag{36}$$

The analytic gradient update, given $z$, for $W_c$ at iteration $k$ with a learning rate $\eta$ is:

$$W_c^{(k+1)} = W_c^{(k)} + \frac{\eta}{\sqrt{M}}\mathcal{F}\left(x - \hat{x}^{(k)}\right) \odot \mathcal{F}(z_{c,t}) \tag{37}$$

Applying the inverse Fourier transform and the cross-correlation theorem, we obtain the effective update rule for the spatial dictionary $D_c'$:

$$D_c'^{(k+1)} = D_c'^{(k)} + \frac{\eta}{M}\left(x - \hat{x}^{(k)}\right) \star z_{c,t} = D_c'^{(k)} + \frac{\eta}{M}\left(x - \frac{1}{\sqrt{M}}\mathcal{F}^{-1}\sum_{i=1}^{C}\mathcal{F}D_i^{(k)} \odot \mathcal{F}z_{i,t}\right) \star z_{c,t} \tag{38}$$

In the lifted scenario, we learn a dictionary $W_L = \{W_{L,c}\}_{c=1}^{C}$ that operates in a higher-dimensional lifted frequency domain. The input signal $x$ is lifted by $L$, and the reconstruction is projected back by $P$. The reconstruction operator is $\hat{x} = P\left(\mathcal{G}_{W_L}(z)\right) = P\left(\frac{1}{\sqrt{M}}\mathcal{F}^{-1}\left(\sum_{c=1}^{C} W_{L,c} \odot \mathcal{F}(z_c)\right)\right)$.

The loss function is defined in the original spatial space:

$$\mathcal{L}(W) = \frac{1}{2}\mathbb{E}_{z\sim\mu}\left[\|x - P\left(\mathcal{G}_{W_L}(z)\right)\|_{\mathcal{X}}^2\right] \tag{39}$$

The gradient of the loss with respect to $W_{L,c}$, given $z$, is:

$$\frac{\partial\mathcal{L}}{\partial W_{L,c}} = -\frac{1}{\sqrt{M}}\mathcal{F}\left(P^\top\left(x - \hat{x}^{(k)}\right)\right) \odot \mathcal{F}(z_c) \tag{40}$$

The analytic gradient update update for $W_{L,c}$ at iteration $k$ with a learning rate $\eta_L$ is:

$$W_{L,c}^{(k+1)} = W_{L,c}^{(k)} + \frac{\eta_L}{\sqrt{M}}\mathcal{F}\left(P^\top\left(x - \hat{x}^{(k)}\right)\right) \odot \mathcal{F}(z_{c,t}) \tag{41}$$

To understand the learning dynamics in the original spatial-domain, we derive the update rule for the effective lifted spatial dictionary $D'_{L,c}$. Taking the inverse Fourier, we get:

$$D'^{(k+1)}_{L,c} = D'^{(k)}_{L,c} + \frac{\eta_L}{M}\left(P^\top\left(x - \hat{x}^{(k)}\right)\right) \star z_{c,t} \tag{42}$$

We also derive the update rule for the effective spatial dictionary in the original space $D'_c$. Left-multiplying the update for $D'_{L,c}$ by $P$, we have:

$$D'^{(k+1)}_c = D'^{(k)}_c + \frac{\eta_L}{M}\left((PP^\top)\left(x - \hat{x}^{(k)}\right)\right) \star z_{c,t} \tag{43}$$

In our specific architecture, we constrain the projection matrix to be the transpose of the lifting matrix such that $P = L^\top$. Substituting this into the update rules yields:

$$W^{(k+1)}_{L,c} = W^{(k)}_{L,c} + \frac{\eta_L}{\sqrt{M}}\mathcal{F}\left(L\left(x - \hat{x}^{(k)}\right)\right) \odot \mathcal{F}(z_{c,t}) \tag{44}$$

$$D'^{(k+1)}_{L,c} = D'^{(k)}_{L,c} + \frac{\eta_L}{M}\left(L\left(x - \hat{x}^{(k)}\right)\right) \star z_{c,t} \tag{45}$$

$$D'^{(k+1)}_c = D'^{(k)}_c + \frac{\eta_L}{M}\left((L^\top L)\left(x - \hat{x}^{(k)}\right)\right) \star z_{c,t} \tag{46}$$

Substituting for $\hat{x}^{(k)}$, the full effective spatial-domain update is:

$$D'^{(k+1)}_c = D'^{(k)}_c + \frac{\eta_L}{M}\left((L^\top L)\left(x - \frac{1}{\sqrt{M}}\mathcal{F}^{-1}\left(\sum_{i=1}^{C}\mathcal{F}(D'^{(k)}_i) \odot \mathcal{F}(z_{i,t})\right)\right)\right) \star z_{c,t} \tag{47}$$

Hence, when $L^\top L = I$, the lifted-SAE-FNO shows equivalent learning dynamics as the standard SAE-FNO for model recovery. ∎

**Proposition D.4** (Architectural Inference Equivalence of SAE-CNN and SAE-FNO). *The architectural inference of an SAE-CNN is equivalent to a sparse autoencoder neural operator (SAE-NO) if the integral operator of SAE-FNO and parameters of SAEs are defined as a convolution where the SAE-FNO's frequency-domain weights $W_c$ are the Fourier transform of SAE-CNN's spatial domain kernels: $W_c = \mathcal{F}D_c$.*

*Proof.* We provide the proof for an encoder that implements a proximal gradient descent type of operation in an unrolled learning setting. For the SAE-CNN, the representation refinement for the $c$-th feature map at every encoder layer follows:

$$z_{c,t+1} = \mathcal{R}\left(z_{c,t} - \alpha\left(\sum_{i=1}^{C}D_i * z_{i,t} - x\right) \star D_c\right) \tag{48}$$

By construction, we incorporate the underlying model from the SAE-FNO to the SAE-CNN: $W_c = \mathcal{F}D_c$, and re-write the gradient of the encoder using Correlation and Convolution Theorems:

$$\mathcal{F}^{-1}\left(\mathcal{F}\left(\sum_{i=1}^{C}D_i * z_{i,t} - x\right) \odot \mathcal{F}D_c^H\right) = \mathcal{F}^{-1}\left(\left(\sum_{i=1}^{C}W_i \odot \mathcal{F}z_{i,t} - \mathcal{F}x\right) \odot (W_c)^H\right) \tag{49}$$

Substituting this back into eq. (48) gives:

$$z_{c,t+1} = \mathcal{R}\left(z_{c,t} - \alpha\mathcal{F}^{-1}\left(\left(\sum_{i=1}^{C}W_i \odot \mathcal{F}z_{i,t}\right) - \mathcal{F}x\right) \odot (W_c)^H\right) \tag{50}$$

This final expression is the SAE-FNO's iterative encoder update for $z_c$ in the spectral domain, where the dictionary is $W$. This completes the proof of architectural inference equivalence. ∎

**Proposition 3.4** (Training Dynamics of SAE-CNN and SAE-FNO). *The training dynamics of the SAE-FNO's frequency-domain weights $\boldsymbol{W}_c^{(k+1)} = \boldsymbol{W}_c^{(k)} + \frac{\eta}{\sqrt{M}}\mathcal{F}\left(\boldsymbol{x} - \hat{\boldsymbol{x}}^{(k)}\right) \odot \mathcal{F}(\boldsymbol{z}_{c,t})$ has an effective update in the original space, expressed as: $\boldsymbol{D}_c^{\prime(k+1)} = \boldsymbol{D}_c^{\prime(k)} + \frac{\eta}{M}\left(\boldsymbol{x} - \hat{\boldsymbol{x}}^{(k)}\right) \star \boldsymbol{z}_{c,t}$. If the architectures satisfy the inference equivalence condition ([Prop. D.4](#), see [Prop. D.5](#) for lifting), then the dynamics are equivalent to that of a SAE-CNN: $\boldsymbol{D}_c^{(k+1)} = \boldsymbol{D}_c^{(k)} + \eta(\boldsymbol{x} - \sum_{i=1}^{C} \boldsymbol{D}_i^{(k)} * \boldsymbol{z}_{i,t}) \star \boldsymbol{z}_{c,t}$.*

*Proof.* For a standard SAE-CNN, we learn a dictionary $\boldsymbol{D}$ by minimizing the reconstruction loss for the data sample $\boldsymbol{x}$. The loss function is:

$$\mathcal{L}(\boldsymbol{D}) = \frac{1}{2}\left\| \boldsymbol{x} - \sum_{c=1}^{C} \boldsymbol{D}_c * \boldsymbol{z}_c \right\|_2^2 \tag{51}$$

The gradient of the loss with respect to the $c$-th kernel $\boldsymbol{D}_c$ is:

$$\frac{\partial \mathcal{L}}{\partial \boldsymbol{D}_c} = -\left( \boldsymbol{x} - \sum_{i=1}^{C} \boldsymbol{D}_i * \boldsymbol{z}_i \right) \star \boldsymbol{z}_c \tag{52}$$

where $\star$ denotes cross-correlation.

The analytic gradient update, given $\boldsymbol{z}$, for $\boldsymbol{D}_c$ at iteration $k$ with a learning rate $\eta$ is:

$$\boldsymbol{D}_c^{(k+1)} = \boldsymbol{D}_c^{(k)} + \eta\left( \boldsymbol{x} - \sum_{i=1}^{C} \boldsymbol{D}_i^{(k)} * \boldsymbol{z}_{i,t} \right) \star \boldsymbol{z}_{c,t} \tag{53}$$

For a standard SAE-FNO, we learn frequency-domain weights $\boldsymbol{W}_c = \mathcal{F}\boldsymbol{D}_c$ by minimizing the reconstruction loss for the data sample $\boldsymbol{x}$. Taking normalization into account, the reconstruction operator is $\hat{\boldsymbol{x}} = \mathcal{G}_{\boldsymbol{W}}(\boldsymbol{z}) = \frac{1}{\sqrt{M}}\mathcal{F}^{-1}\left( \sum_{c=1}^{C} \boldsymbol{W}_c \cdot (\mathcal{F}\boldsymbol{z}_c) \right)$, where $\boldsymbol{W}_c = \mathcal{F}\boldsymbol{D}_c$, where $M$ is the number of modes. The loss function is:

$$\mathcal{L}(\boldsymbol{W}) = \frac{1}{2}\mathbb{E}_{\boldsymbol{z}\sim\mu}\left[ \|\boldsymbol{x} - \mathcal{G}_{\boldsymbol{W}}(\boldsymbol{z})\|_{\mathcal{X}}^2 \right] \tag{54}$$

where $\boldsymbol{x} = \mathcal{G}_{\boldsymbol{D}^*}(\boldsymbol{z})$ is the ground-truth signal.

The gradient of the loss with respect to the $c$-th frequency-domain kernel $\boldsymbol{W}_c$ is:

$$\frac{\partial \mathcal{L}}{\partial \boldsymbol{W}_c} = -\frac{1}{\sqrt{M}}\mathcal{F}\left(\boldsymbol{x} - \hat{\boldsymbol{x}}\right) \odot \mathcal{F}(\boldsymbol{z}_c) \tag{55}$$

The analytic gradient update, given $\boldsymbol{z}$, for $\boldsymbol{W}_c$ at iteration $k$ with a learning rate $\eta$ is:

$$\boldsymbol{W}_c^{(k+1)} = \boldsymbol{W}_c^{(k)} + \frac{\eta}{\sqrt{M}}\mathcal{F}\left(\boldsymbol{x} - \hat{\boldsymbol{x}}^{(k)}\right) \odot \mathcal{F}(\boldsymbol{z}_{c,t}) \tag{56}$$

Applying the inverse Fourier transform and the cross-correlation theorem, we obtain the effective update rule for the spatial dictionary $\boldsymbol{D}_c'$:

$$\boldsymbol{D}_c^{\prime(k+1)} = \boldsymbol{D}_c^{\prime(k)} + \frac{\eta}{M}\left(\boldsymbol{x} - \hat{\boldsymbol{x}}^{(k)}\right) \star \boldsymbol{z}_{c,t} \tag{57}$$

This is equivalent to the gradient update of SAE-CNN with a scaling factor of $\frac{1}{M}$ on the learning rate, which arises from the inverse Fourier normalization conventions. ∎

**Proposition D.5** (Architectural Inference Equivalence of L-SAE-CNN and L-SAE-FNO). *The architectural inference of a L-SAE-CNN is equivalent to a L-SAE-FNO if the integral operator of L-SAE-FNO and parameters of SAEs are defined as a convolution where the L-SAE-FNO's lifted frequency-domain weights $\boldsymbol{W}_{L,c}$ are the Fourier transform of L-SAE-CNN's lifted spatial domain kernels: $\boldsymbol{W}_{L,c} = \mathcal{F}\boldsymbol{D}_{L,c}$.*

*Proof.* We provide proof for an encoder that implements a proximal gradient descent type of operation in an unrolled learning setting. For the L-SAE-CNN, the representation refinement for the $c$-th feature map at every encoder layer follows:

$$z_{c,t+1} = \mathcal{R}\left(z_{c,t} - \alpha\left(\sum_{i=1}^{C} D_{L,i} * z_{i,t} - Lx\right) \star D_{L,c}\right) \tag{58}$$

By construction, we incorporate the underlying model from L-SAE-FNO to L-SAE-CNN: $W_{L,c} = \mathcal{F}D_{L,c}$, and re-write the gradient of the encoder using Correlation and Convolution Theorems:

$$\mathcal{F}^{-1}\left(\mathcal{F}\left(\sum_{i=1}^{C} D_{L,i}^{(k)} * z_{i,t} - Lx\right) \odot \mathcal{F}D_{L,c}^{H}\right) = \mathcal{F}^{-1}\left(\sum_{i=1}^{C} W_{L,i}^{(k)} \odot \mathcal{F}z_{i,t} - \mathcal{F}(Lx)\right) \odot (W_{L,c})^{H} \tag{59}$$

Substituting this back into eq. (58) gives:

$$z_{c,t+1} = \mathcal{R}\left(z_{c,t} - \alpha\mathcal{F}^{-1}\left[\left(\left(\sum_{i=1}^{C} W_{L,i} \odot \mathcal{F}(z_{i,t})\right) - \mathcal{F}(Lx)\right) \odot (W_{L,c})^{H}\right]\right) \tag{60}$$

This final expression is the L-SAE-FNO's iterative update for $z_c$ in the lifted spectral domain, where the dictionary is $W_{L,c}$. This completes the proof of architectural inference equivalence. ∎

**Proposition D.6** (Architectural Inference Equivalence of Lifting in SAE-FNO). *The architectural inference of an SAE-FNO is equivalent to a Lifted SAE-FNO (L-SAE-FNO) if the projection is tied to the lifting operator ($P = L^{\top}$) that is orthogonal ($L^{\top}L = I$), i.e., under the same learned model $W_c = PW_{L,c}$, the representation dynamics and output of the networks are equivalent.*

*Proof.* We prove this equivalence by establishing a chain of architectural inference equivalences that connects the SAE-FNO to the Lifted SAE-FNO, leveraging the propositions previously established:

1. From Proposition D.4, the architectural inference of an SAE-FNO is equivalent to SAE-CNN.

2. From Proposition D.1, the architectural inference of an SAE-CNN is equivalent to L-SAE-CNN.

3. From Proposition D.5, the architectural inference of a L-SAE-CNN is equivalent to L-SAE-FNO.

By the transitive property of these equivalences, we can establish a direct architectural inference equivalence between SAE-FNO and L-SAE-FNO under the same lifting-projection conditions. ∎

