# OpenReview forum: "Sparse Autoencoder Neural Operators: Model Recovery in Function Spaces"
_NeurIPS.cc/2025/Workshop/UniReps — UniReps2025 oral_

### Official Review · Reviewer_KgrK · 2025-09-13
**Interesting application of SAEs to infinite-dimensional function spaces**

**Confidence:** 2

**Review:**

## Summary
The paper extends sparse autoencoders (SAEs) from finite-dimensional Euclidean data to infinite-dimensional function spaces, introducing "Sparse Autoencoder Neural Operators" (SAE-NOs). Motivated by the Platonic Representation Hypothesis (different architectures converge to similar latent structures), the authors ask whether SAEs and neural operators recover the same underlying concepts. They introde two key ideas:
- Lifting: mapping inputs into a higher-dimensional space to act as a preconditioner and speed convergence.
- SAE-NO / L-SAE-NO: neural-operator-based autoencoders for learning sparse dictionaries over function spaces.

### Strengths
- Clear motivation bridging two active areas: mechanistic interpretability via SAEs and neural operators for PDEs.
- Neural-operator SAEs have an inductive bias toward recovering smooth concepts and remain stable under changes of resolution, whereas standard SAEs degrade.

### Weaknesses
- All results are on synthetic sparse generative data only
- Comparisons with other methods are missing.
- Figures captions needs to be improved, in the current state they are pretty uninformative.

**Score:**

3

**Topic Fit:**

3

---

### Official Review · Reviewer_DEor · 2025-09-15
**Very promising work but limited to synthetic datasets, very good fit for the workshop**

**Confidence:** 3

**Review:**

## Summary
This extended abstract introduces **Sparse Autoencoder Neural Operators (SAE-NO)**, an extension of sparse autoencoders (SAEs) to function spaces, aimed at improving mechanistic interpretability of large neural operators. Motivated by the **Platonic Representation Hypothesis**, the work explores conditions under which representations converge across architectures. Key contributions include:
- Formulating sparse model recovery in lifted and infinite-dimensional function spaces.
- Introducing **lifted SAEs (L-SAE)** and **SAE-NOs** with inductive biases (e.g., lifting as a preconditioner, Fourier domain operators).
- Demonstrating that SAE-NOs can recover smooth concepts more robustly, generalize across resolutions, and reduce to classical SAE dynamics under certain conditions.
Experiments are conducted on synthetic data, comparing SAEs, L-SAEs, and SAE-NOs.


## Strengths
1. **Novelty & Scope**: Extends exiting SAE framework to Neural operators providing a novel way to computing them.
2. **Clarity of Motivation**: Well-motivated by the Platonic Representation Hypothesis
3. **Technical Contributions**: Provides theoretical formulations, propositions about training dynamics, and connections between SAEs, Fourier operators, and convolutional settings.

I believe this work is in the right track and am very much impressed with the work. Unfortunately I wont be able to provide too much of a feedback or improvements that can be done.

**Score:**

5

**Topic Fit:**

3

---

### Official Review · Reviewer_jzWo · 2025-09-16
**Good Work: Substantial Extension of SAEs to Function Spaces**

**Confidence:** 4

**Review:**

## Overall Evaluation

This paper makes a **substantial extension of sparse autoencoders (SAEs) into function spaces**, formulating _Lifted SAEs (L-SAE)_ and _SAE Neural Operators (SAE-NO)_. The approach leverages **Fourier domain formulations** and **operator learning** to study sparse model recovery in infinite-dimensional settings. The work is conceptually strong, mathematically well-grounded, and yields intriguing empirical results — particularly that L-SAE acts as a preconditioner to accelerate training and that SAE-NOs robustly recover models under varying resolutions.

The main limitation is stylistic: the paper is **significantly shortened to fit the short version**, which disrupts the flow and makes it difficult to follow in places. Nonetheless, the contribution is strong enough to merit acceptance.

## Quality and Clarity

**Quality**: The paper demonstrates solid overall quality, with a well-balanced integration of theoretical analysis and empirical validation.

**Clarity**: The flow is hampered by the compressed format. Several definitions (e.g., lifted spaces, Fourier integral operators) appear with minimal motivation, making the exposition heavy. Expanding these sections would help readability.

## Originality

**Substantial extension of SAE**: Extending sparse autoencoders to neural operators and function spaces is a novel and non-trivial contribution. It bridges sparse interpretability with operator learning.

**Fourier-based formulation**: Using Fourier neural operators (FNOs) to define SAE-FNO and L-SAE-FNO is particularly original and well motivated by the equivalence between convolution and Fourier multiplication.

**New perspective on training dynamics**: Showing that lifting acts as a preconditioner and speeds convergence is both elegant and practically valuable.

## Significance

**Practical significance**: Results show that L-SAE speeds up training and SAE-FNOs can robustly recover models even under resolution shifts. These findings have implications for scientific machine learning and mechanistic interpretability of large neural operators.

**Theoretical significance**: The equivalence proofs connecting SAE-CNN, SAE-FNO, and L-SAE variants provide a rigorous foundation for studying interpretability in operator learning.

**Limitation**: The shortened format hurts readability.

## Pros and Cons

1.  (+) **Substantial extension**: Extends SAEs beyond Euclidean space into function spaces, which is both ambitious and impactful.
1.  (+) **Novel Fourier-based approach**: Introduces Fourier transformations and neural operators into sparse model recovery.
1.  (+) **Intriguing results**: L-SAE accelerates training; SAE-NOs generalize across resolutions where SAEs fail, and both recover models superbly.
1.  (-) Overly compressed presentation

**Score:**

4

**Topic Fit:**

2